# Natural selection in the evolution of SARS-CoV-2 in bats created a generalist virus and highly capable human pathogen

Oscar A. MacLean[1], Spyros Lytras[1], Steven Weaver[2], Joshua B. Singer[1], Maciej F. Boni[3], Philippe Lemey[4], Sergei L. Kosakovsky Pond[2]*, David L. Robertson[1]*

**1** MRC-University of Glasgow Centre for Virus Research, Glasgow, United Kingdom, **2** Temple University, Institute for Genomics and Evolutionary Medicine, Philadelphia, Pennsylvania, United States of America, **3** Center for Infectious Disease Dynamics, Department of Biology, Pennsylvania State University, University Park, Pennsylvania, United States of America, **4** Department of Microbiology, Immunology and Transplantation, Rega Institute, KU Leuven, Leuven, Belgium

☯ These authors contributed equally to this work.

* spond@temple.edu (SLKP); david.l.robertson@glasgow.ac.uk (DLR)

**Data Availability Statement:** All relevant data are within the paper, its Supporting Information files or associated online resources.

## Abstract

Virus host shifts are generally associated with novel adaptations to exploit the cells of the new host species optimally. Surprisingly, Severe Acute Respiratory Syndrome Coronavirus 2 (SARS-CoV-2) has apparently required little to no significant adaptation to humans since the start of the Coronavirus Disease 2019 (COVID-19) pandemic and to October 2020. Here we assess the types of natural selection taking place in *Sarbecoviruses* in horseshoe bats versus the early SARS-CoV-2 evolution in humans. While there is moderate evidence of diversifying positive selection in SARS-CoV-2 in humans, it is limited to the early phase of the pandemic, and purifying selection is much weaker in SARS-CoV-2 than in related bat *Sarbecoviruses*. In contrast, our analysis detects evidence for significant positive episodic diversifying selection acting at the base of the bat virus lineage SARS-CoV-2 emerged from, accompanied by an adaptive depletion in CpG composition presumed to be linked to the action of antiviral mechanisms in these ancestral bat hosts. The closest bat virus to SARS-CoV-2, RmYN02 (sharing an ancestor about 1976), is a recombinant with a structure that includes differential CpG content in Spike; clear evidence of coinfection and evolution in bats without involvement of other species. While an undiscovered "facilitating" intermediate species cannot be discounted, collectively, our results support the progenitor of SARS-CoV-2 being capable of efficient human–human transmission as a consequence of its adaptive evolutionary history in bats, not humans, which created a relatively generalist virus.

## Introduction

In December 2019, an outbreak of pneumonia cases in the city of Wuhan, China, was linked to a novel coronavirus. Evolutionary analysis identified this new virus to humans as a severe acute respiratory syndrome-related virus [1], in the *Sarbecovirus* subgenus of the *Betacoronavirus*

 

**Funding:** DLR is funded by the Medical Research Council (MC_UU_1201412) and Wellcome Trust (220977/Z/20/Z). OAM is funded by the Wellcome Trust (206369/Z/17/Z). SLKP and SW are supported in part by the National Institutes of Health (R01 AI134384 (NIH/NIAID)) and the National Science Foundation (award 2027196). PL acknowledges funding from the European Research Council under the European Union's Horizon 2020 research and innovation programme (grant agreement no. 725422-ReservoirDOCS), the European Union's Horizon 2020 project MOOD (874850), the Wellcome Trust through project 206298/Z/17/Z (The Artic Network) and the Research Foundation – Flanders ('Fonds voor Wetenschappelijk Onderzoek – Vlaanderen', G066215N, G0D5117N and G0B9317N). MFB is funded by a grant from the Bill and Melinda Gates Foundation (INV-005517) and by NIH/NIAID Center of Excellence in Influenza Research and Surveillance contract (HHS N272201400007C). The funders had no role in study design, data collection and analysis, decision to publish, or preparation of the manuscript.

**Competing interests:** The authors have declared that no competing interests exist.

**Abbreviations:** aBSREL, adaptive branch-site random effects likelihood; COVID-19, Coronavirus Disease 201; CV, coefficient of variation; dN, nonsynonymous substitution rate; dS, synonymous substitution rate; EM, expectation–maximization; FEL, fixed effects likelihood; GTR, general time-reversible; hACE2, human angiotensin-converting enzyme 2; HMM, hidden Markov model; MCC, maximum clade credibility; MCMC, Markov chain Monte Carlo; MEME, mixed effects model of evolution; MERS, Middle East Respiratory Syndrome; nCoV, new coronavirus; ORF, open reading frame; OU, Ornstein–Uhlenbeck; QC, quality control; RDA, relative dinucleotide abundance; RdRp, RNA-dependent RNA polymerase; SARS-CoV-2, Severe Acute Respiratory Syndrome Coronavirus 2; SDUc, corrected synonymous dinucleotide usage; SLAC, single-likelihood ancestor counting; ZAP, Zinc finger Antiviral Protein.

genus, sister lineage to the original SARS virus; subsequently named Severe Acute Respiratory Syndrome Coronavirus 2 (SARS-CoV-2) to reflect this relationship to SARS-CoV [2]. This novel lineage represents the seventh known human-infecting member of the *Coronaviridae*. The initial outbreak of human cases of the virus was connected to the Huanan Seafood Wholesale Market in Wuhan [3], and while related viruses have been found in horseshoe bats [4] and pangolins [5], their divergence represents decades of evolution [6] leaving the direct origin of the pandemic unknown. In addition to elucidating the transmission route from animals to humans, key questions for assessing future risk of emergence are: (i) what is the extent of evolution, if any, required for a bat virus to transmit to humans, and (ii) what subsequent evolution will occur once the virus is established within the human population?

The first SARS virus outbreak in 2002/2003, causing approximately 8,000 infections, and its reemergence in late 2003, causing 4 infections, were linked to Himalayan palm civets and raccoon dogs in marketplaces in Guangdong Province [7,8]. Later, it became clear that while these animals may have been conduits for spillover to humans, they were not true viral reservoirs [9]. Extensive surveillance work subsequently identified related viruses circulating in horseshoe bats in China, some of which can replicate in human cells [10,11]. The bat viruses most closely related to SARS-CoV (hereafter referred to as SARS-CoV-1 for clarity), can bind to human angiotensin-converting enzyme (hACE2, the receptor SARS-CoV-2 also uses for cell entry), while the addition of a host protease is required to cleave the Spike protein before it can bind hACE2 for the more divergent bat viruses tested (S1 Fig) [12]. Two of the key changes which appear to have been required to generate this highly capable pathogen, the specific receptor binding domain sequence and the inserted furin cleavage site, can all be traced to bat coronaviruses [6,13–15]. Collectively, these results demonstrate that, unlike most other RNA viruses which acquire adaptations after switching to a new host species [16,17] for efficient replication and spreading as successfully as exhibited by SARS-CoV-2, the *Sarbecoviruses*—which already transmit frequently among bat species [18]—can exploit the generalist properties of their ACE2 binding ability, facilitating successful infection of non-bat species, including humans. A main difference between SARS-CoV-1 and SARS-CoV-2 is the increased binding affinity for hACE2 in the latter [19] permitting more efficient use of human cells and the upper respiratory tract, and on average lower severity but, paradoxically—due to the higher number of infections—higher disease burden.

There is intense interest in the mutations emerging in the SARS-CoV-2 pandemic [20–22]. Although the vast majority of observed genomic change is expected to be "neutral" [23,24], mutations with functional significance to the virus will likely arise, as they have in many other viral epidemics and pandemics [25]. In SARS-CoV-2, amino acid replacements in the Spike protein may reduce the efficacy of vaccines, replacements in proteases and polymerases will result in acquired drug resistance, and other mutations could change the biology of the virus, e.g., enhancing its transmissibility as demonstrated for the replacement D614G [26], contributing to adaption to humans, a new host species.

A main way to begin to understand the functional impact of mutations is to characterise the selective regime they are under. Mutations which are under positive selection are of particular interest as they are more likely to reflect a functional change. However, identifying mutations under positive selection from frequency data alone can be misleading, as allele frequencies in viral pandemics are significantly driven by biased sampling, founder effects, and superspreading events [22]. Exponentially growing populations can increase in average fitness [27]; however, they are also expected to exhibit elevated genetic drift, with deleterious mutations "surfing" expansion waves [28].

Here, we investigate the evolutionary history of bat *Sarbecoviruses* that shaped the emergence and rapid spread of SARS-CoV-2 by contrasting results from comprehensive searches

 

for signatures of positive selection (a measure of molecular adaptation) in the virus circulating in humans since the Coronavirus Disease 2019 (COVID-19) outbreak began, to signatures of historic selection acting on related bat viruses. We use an array of selection detection methods (summarised in S2 Fig) on a set of (i) 133,741 SARS-CoV-2 genomes sampled from December 2019 to October 2020 and on (ii) 69 *Sarbecovirus* genomes (including a representative of SARS-CoV-1 and SARS-CoV-2 sequences) separated into phylogenetically congruent regions based on detected recombination patterns [6]. Finally, we detect a shift in CpG representation on the *Sarbecovirus* tree, associated with the phylogenetic clade SARS-CoV-2 is found in.

## Results

### Evidence of nearly neutral evolution and relatively weak purifying selection in the first 11 months of the SARS-CoV-2 pandemic

We first analyse selection acting on the encoded amino acids of SARS-CoV-2 using 133,741 quality control (QC)-filtered genome sequences from the GISAID database as of 12 October 2020, representing a sample of the variants circulating in humans during the first 11 months of the pandemic (see Methods, S5 Table). Purifying selection acts more strongly on nonsynonymous sites, supported by the estimate that the nonsynonymous substitution rate (dN) was only 4% the speed of the synonymous substitution rate (dS) in the divergence between SARS--CoV-2 and the bat *Sarbecovirus* RaTG13 [29]. We used the phylogenetically corrected single-likelihood ancestor counting (SLAC) method [30] to calculate "frequency-stratified" dN/dS estimates. This qualitative analysis revealed that, on average, the evolutionary process is indicative of weakly purifying selection, with point estimates of dN/dS ranging from 0.4 to 0.9, and dN/dS generally declining with increasing frequency (Fig 1A). Most of the bootstrap-based 95% confidence intervals exclude a dN/dS = 1, although for ranges with a small number of variants (e.g., ≥5,000 sequences), there exists a high degree of uncertainty in these estimates. The vast majority of 20,687 observed mutations occur at very low frequency, with 79% of mutations observed in 10 or fewer of the 133,741 SARS-CoV-2 genome sequences analysed (Fig 1A), consistent with a model of exponential population growth of virus spread (S1 Text).

Our initial searches for signatures of departures from neutral evolution in the SARS-CoV-2 sequence data from late March produced numerous false positive signals, due to artefactual lab recombination and other sequencing errors found on terminal branches (see S2 Text and S1 Table). Subsequently, we have been performing regular analyses of the data excluding these terminal branches (available online at http://covid19.datamonkey.org). As more sequences were deposited in public databases (Fig 1B), we expected to see increasingly more variation, with >95% of codons in Spike (S) and RNA-dependent RNA polymerase (RdRp) containing some variation in most recent analyses (Fig 1C). Many of those variants may be due to sequencing errors, but the rapid increase of variants is clear in the sequence data sampled from the circulating viruses. Similar increasing trends are seen for sites with amino-acid variants that are present in at least 2 sequences (47.1% in S and 36.5% in RdRp). Interestingly, the number of unique sequences has increased more slowly than the number of sequenced genomes (Fig 1B), and the fraction of sites inferred to be under positive selection remains approximately 2% in both genes, while the fraction of sites detected to be under purifying selection is up to 11.4% in S and 7.2% in RdRp. This observation shows SARS-CoV-2 is evolving relatively slowly with no dramatic increases in selective pressures occurring over the sampling period December 2019 to October 2020. This pattern is further confirmed by the relatively stable behaviour of gene-wide dN/dS estimates (Fig 1D), both on internal branches, which include successful transmissions, and terminal branches, which sometimes include intrahost "dead-ends" or deleterious variation, and thus have higher estimates [31].

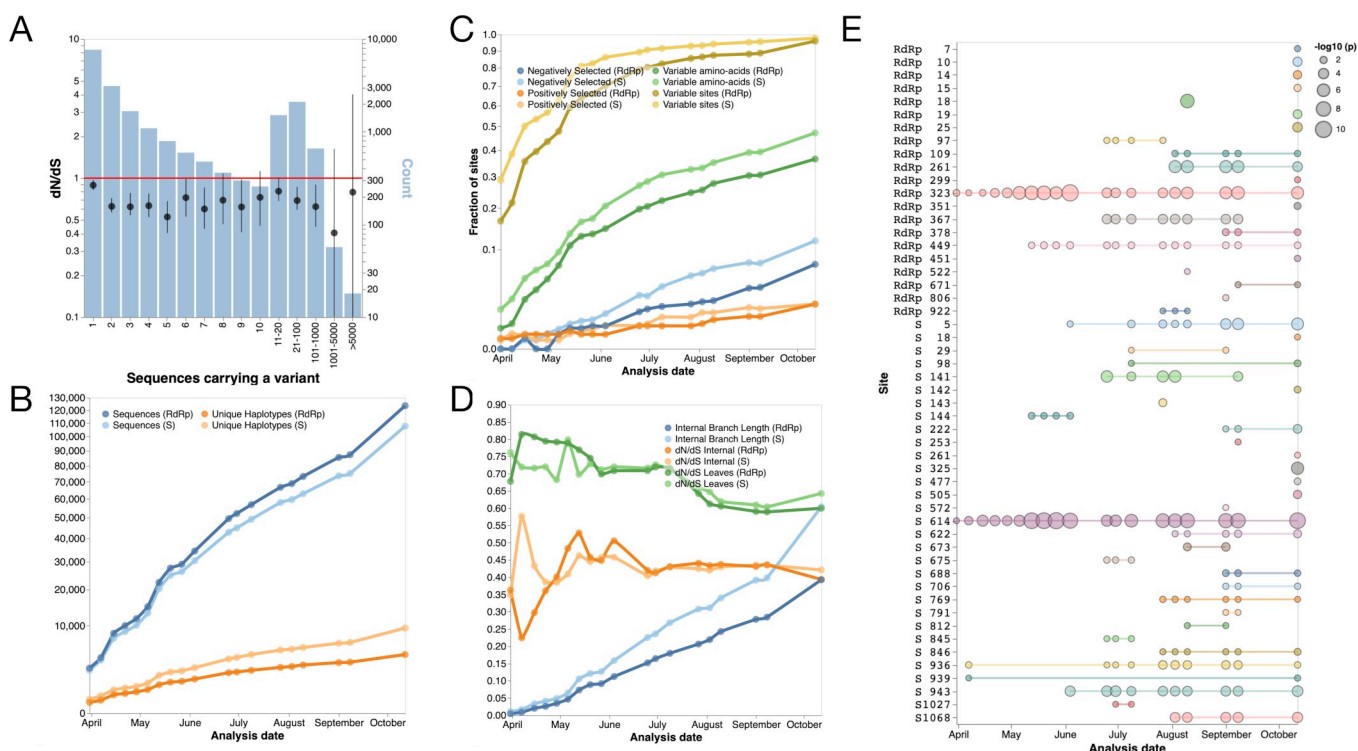

**Fig 1.** **(A)** Estimates of molecular adaptation (dN/dS) for 133,741 SARS-CoV-2 genome sequences based on the counting SLAC method [30] (black circles—point estimates, error-bars: 95% interpercentile range based on 500 bootstrap replicates) and the number of variants as a function of their frequency (blue bars). **(B)** Cumulative number of Spike and RdRp gene sequences in GISAID passing QC filters from 31 March to 12 October 2020, and the number unique haplotypes among them. **(C)** Cumulative fractions of codon sites in Spike and RdRp, which harbour different types of sequence variants, are positively selected (MEME [36] $p \leq 0.05$, internal branches only), or are negatively selected (FEL [30] $p \leq 0.05$, internal branches only). **(D)** Estimates of gene-wide dN/dS in Spike and RdRp on internal branches and terminal branches (MG94xREV model), and the total length of internal tree branches, which serves as a good proxy of statistical power to detect selection. And **(E)** statistical evidence for episodic positive selection at 52 codons in SARS-CoV-2 Spike (S) and RdRp that reached significance ($p \leq 0.05$) at least once during the analysis period. The list of accessions for the SARS-CoV-2 sequences downloaded from GISAID on 12 October 2020 are provided in S5 Table. dN, nonsynonymous substitution rate; dS, synonymous substitution rate; FEL, fixed effects likelihood; MEME, mixed effects model of evolution; QC, quality control; RdRp, RNA-dependent RNA polymerase; S, Spike; SARS-CoV-2, Severe Acute Respiratory Syndrome Coronavirus 2; SLAC, single-likelihood ancestor counting.

Even in Spike, which is being assiduously scrutinized for selection due to its immunogenic and phenotypic importance, overall selective pressure is stable over time and consistent with weak purifying selection. The genetic homogeneity of SARS-CoV-2 results in very shallow phylogenetic trees, despite >130,000 collected sequences, with cumulative branch lengths only about 0.6 (S) and 0.4 (RdRp) substitutions/site (Fig 1D), somewhat limiting the statistical power of comparative methods [32]. There are individual sites in the SARS-CoV-2 genomes which possess statistically detectable signatures of episodic diversifying selection along internal tree branches. In selection scans of Spike and RdRp, for example, there are 4 sites that appear to be selected in data prior to April 15 (early pandemic, Fig 1E). These include Spike 614 and RdRp 323 which are in nearly perfect linkage disequilibrium ($R^2 > 0.99$), with the former extensively discussed by others as having some functional significance [33,34], while the latter has received much less attention. The majority of these flagged sites, however, are not detected until later in the pandemic, suggesting that they were not critical for the exponential spread of the virus. Most observed mutations are only transient in the viral population. This is because they are mildly deleterious and will fail to persist over longer time periods of the viral outbreak, being gradually purged by purifying selection [31,35]. Exponential growth, consistent with the

SARS-CoV-2 trajectory (S1 Text), is known to reduce the efficacy of purifying selection [28], with deleterious mutations able to "surf" expansion waves. It is likely that many of these putatively deleterious segregating mutations will ultimately be lost, reducing the long-term dN/dS, though their persistence will be influenced by future demographic patterns [28], and the effect of lockdowns on the circulating variants. With increasing levels of host immunity helped by the deployment vaccines and ongoing widespread SARS-CoV-2 circulation, we fully expect to see increased evidence for adaptive evolution in Spike and other genes, but this evolution is not relevant to initial adaptation for efficient spread among human hosts. This can already be seen in the data, with 15/52 sites appearing on the list of positively selected sites only in the 12 October 2020 (most recent) analysis (Fig 1E).

## What about the bats? Evidence for positive selection in *Sarbecoviruses*

Coronaviruses frequently recombine in their bat hosts, with the Spike open reading frame (ORF) being an apparent hotspot for recombination events, with potentially adaptive implications for the viruses, e.g., antigenic shift in the context of immune evasion [9,37–40]. To avoid the confounding effects of recombination on the inference of selection patterns, we separately analysed maximally nonrecombinant regions, as derived in Boni and colleagues [6] (see Methods). For each of the 19 nonrecombinant ORF regions identified, we define as a working nomenclature the new coronavirus "nCoV" clade, the set of viruses closest to SARS-CoV-2 lineage in the *Sarbecovirus* phylogeny (see Methods). We find that genomic sites are generally subject to stabilising selection in this nCoV clade, with 8,184/9,744 (84%) of codon sites conserved at the amino-acid level, and 4,274 (43.7%) sites, of which 3,388 were variable at the nucleotide level, showing evidence of purifying selection in this lineage (using the fixed effects likelihood (FEL) method [30], Fig 2).

To search for evidence of positive selection on specific branches of the phylogeny in the nonrecombinant regions, we first used the adaptive branch-site random effects likelihood (aBSREL) method [41]. Our analysis finds that diversifying selection left its imprints primarily in the deepest branches of the nCoV clade or lineage leading to it, with no evidence of selection in the terminal branch leading to SARS-CoV-2 (Fig 2). This is consistent with the nonhuman progenitor of SARS-CoV-2 requiring little or no novel adaptation to successfully infect humans. Still, no model can detect all signatures of historic genomic adaptation, and mutations which may enable SARS-CoV-2 to infect humans could have arisen by genetic drift in the reservoir host before human exposure.

We next sought evidence of ORF-specific positive selection across the full phylogeny of the nCoV clade (episodic diversifying selection) using BUSTED[S] [42], coupled with a hidden Markov model (HMM) with 3 rate categories to describe site-specific synonymous rate variation and allow autocorrelation across neighbouring sites in these rates [43]. Six nonrecombinant regions of Orf1ab, Spike, and ORF N show evidence of episodic diversifying positive selection in the nCoV clade (Fig 2A). This finding is consistent with evidence of positive selection operating on Orf1ab in Middle East Respiratory Syndrome (MERS) [44], and Spike and N proteins being essential for immune recognition. All segments show a nontrivial signal of synonymous rate site-to-site variation, highlighting the importance of accounting for synonymous rate variation in this study, as recent literature [45] indicates that when the coefficient of variation (CV) for the distribution of site-to-site synonymous rates exceeds 0.5. Importantly, not accounting for synonymous rate variation risks mistaking it for evidence of selection.

A subsequent scan for the specific codon sites showing evidence of nCoV-specific selection using the mixed effects model of evolution (MEME) method [36] revealed 85 individual sites inferred to evolve subject to episodic diversifying selection in the nCoV clade or lineage

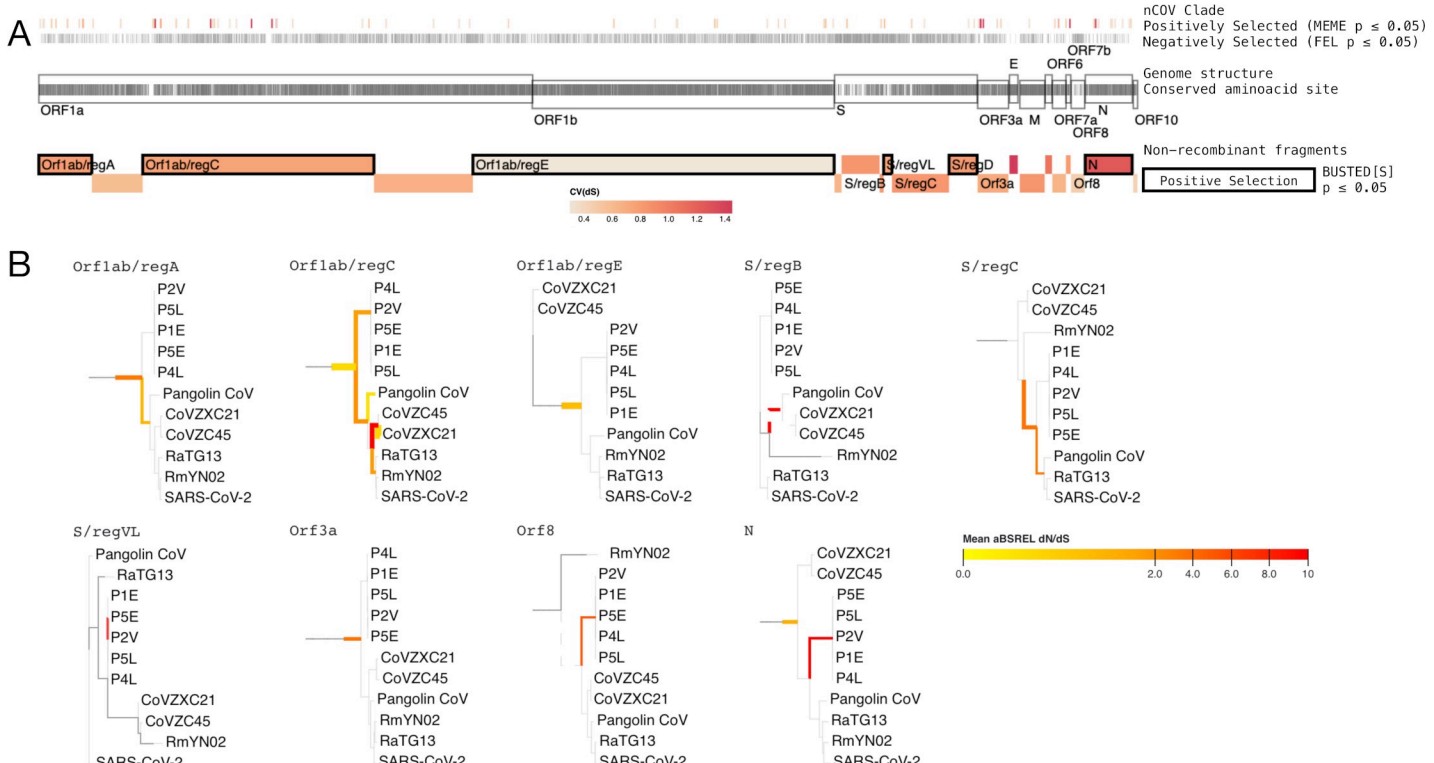

**Fig 2. Schematic of the nonrecombinant ORF regions used for the nCoV clade selection analyses.** **(A)** Displays (top to bottom): Individual codon sites ($N = 85$), mapped to SARS-CoV-2 genomic coordinates, found to be under episodic positive selection in the nCoV clade (MEME $p \leq 0.05$); brighter colours indicate that a larger fraction of lineages was subject to selection; sites ($N = 3,388$) subject to negative selection in the nCoV clade (FEL $p \leq 0.05$); genome structure of SARS-CoV-2; ticks inside the genome structure indicate sites that are conserved in the nCoV clades ($N = 8,184$); nonrecombinant fragments ($N = 20$) found in the Boni and colleagues [6] analysis; colours show the coefficient of variation for the distribution of site-level synonymous rates. **(B)** The nCoV clade for 9/20 nonrecombinant segments that exhibit any evidence of branch-level selection according to the aBSREL method. Branches with significant tests ($p \leq 0.05$) are shown in the orange-red colours; the colour is based on the average dN/dS estimate for these branches, thickness is proportional to the number of individual sites (genome-wide) that have evidence for positive selection along that branch. The list of GenBank and GISAID accessions for the *Sarbecovirus* sequences used are provided in S4 Table. aBSREL, adaptive branch-site random effects likelihood; dN, nonsynonymous substitution rate; dS, synonymous substitution rate; FEL, fixed effects likelihood; MEME, mixed effects model of evolution; nCoV, new coronavirus; ORF, open reading frame; SARS-CoV-2, Severe Acute Respiratory Syndrome Coronavirus 2.

leading to it (Fig 2B and S2 Table). Most of these sites (55/85) are found on Orf1ab, which is also substantially longer than the other regions. Consistent with BUSTED[S] results, Spike and ORF N have the next greatest number of selected sites (7 and 11, respectively; S2 Table). Interestingly, despite its short length, ORF3a has 6 sites with evidence of positive selection. Little is known about the function of this accessory gene; however, it could be related to immune evasion, like that of other short accessory genes (e.g., Orf3b) [46]. Still, this analysis does not allow attributing specific functional relevance to the individual sites.

The BUSTED[S] method also partitioned synonymous rate variation into 3 rate classes across the sites. The majority of regions showed large, in some cases more than 20-fold, differences between rate classes, with all 3 classes representing a substantial proportion of sites for most regions (S3 Fig), with varying degrees of autocorrelation. This suggests that strong purifying selection is acting on some synonymous sites (e.g., due to conserved RNA motifs, RNA structures, or overlapping ORFs), and some synonymous mutations in the SARS-CoV-2 genome may not be selectively neutral or occur at sites that are hypervariable. Some synonymous rate variation may also be attributed to the 5′ and 3′ context-specific mutation rate variation observed in SARS-CoV-2 [47]. Because multinucleotide mutations could be taking place in SARS-CoV-2 genomes, e.g., trinucleotide substitutions in N at positions 28881–28883 [48]),

and not considering them in models can lead to false positives in certain cases [49], we modified BUSTED[S] to include them and to estimate the extent of multinucleotide mutations in the SARS-CoV-2 data [45]. For 11 out of 20 segments (S3 Table), this model provides a better information theoretical fit to the data, including 2 of 5 segments where BUSTED[S] found evidence of positive selection (N and ORF1ab/regE). For N, both models agree that there is evidence of positive selection, while for ORF1ab/regE, the inclusion of multiple nucleotide changes removes the signal of selection due to their effect (S3 Table). Overall, the addition of multinucleotide mutations does not dramatically alter the overall selective picture.

## Patterns of CpG depletion and recombination in the nCoV clade

Genome composition measures, such as dinucleotide representation and codon usage, can also be an informative tool for characterising the host evolutionary history of a virus [50]. Two major host antiviral mechanisms are currently thought to drive the depletion of CpG dinucleotides (a cytosine followed by a guanine in the 5′ to 3′ direction) in virus genomes. This depletion is apparently primarily mediated either through selective pressures by a CpG-targeting mechanism involving the Zinc finger Antiviral Protein (ZAP) [51] or C to U hypermutation by APOBEC3 cytidine deaminases [52], and it has been demonstrated that the CpG-binding ZAP protein inhibits SARS-CoV-2 replication in human lung cells [53]. These evolutionary forces are likely to vary across tissues and between hosts [50]. Thus, a smaller or greater level of CpG depletion in specific viral lineages may be indicative of a switch in the evolutionary environment of that lineage or its ancestors, although care must be taken to not overinterpret such results [54].

We examined the CpG representation using the corrected Synonymous Dinucleotide Usage (SDUc) framework, controlling for amino acid abundance and single nucleotide composition bias in the sequences [55]. In Orf1ab, the longest ORF in the genome, all viruses show CpG underrepresentation (SDUc < 1), while viruses in the nCoV clade have even lower CpG levels than the other *Sarbecoviruses* (S4 Fig). To further examine this trend, we fitted CpG representation as traits in a phylogenetic comparative approach [56], using the 2 longest putatively nonrecombinant regions in the genome alignment: NRR1, comprising of subregions of Orf1ab regC, regD, and regE, and NRR2, identical to Orf1ab regE apart from a small region at one end being removed based on RmYN02's recombination pattern (see Methods). This method formally tests for adaptive shifts of quantitative traits across a phylogeny and estimates where these events have taken place on the tree. In this way, we identified an adaptive shift favouring CpG suppression in the lineage leading to the nCoV clade (Fig 3A and 3B). This CpG shift may indicate a change of evolutionary environment, e.g., host or tissue preference, since CpG depletion following host switches has been observed in other human infecting RNA viruses, such as Influenza B [50].

We investigated if this adaptive shift in CpG content at the base of the nCoV clade was associated with an elevation in substitution rate, which would be expected if dinucleotide mutations were being driven to fixation by positive selection. We tested a model which allowed for a rate change on the internal branch connecting the nCoV clade to the ancestral node and found significant evidence of an elevated substitution rate, consistent with an adaptive shift along this branch (Fig 3A). We could not distinguish between a transient acceleration on the internal branch or an nCoV clade-wide substitution rate acceleration (S3 Text). However, we propose that the elevated rate specific to the nCoV clade ancestral lineage may explain some of the variation between different *Coronaviridae* substitution rates, previously attributed to a time-dependent evolutionary rate phenomenon [6]. Future estimates of split timings should

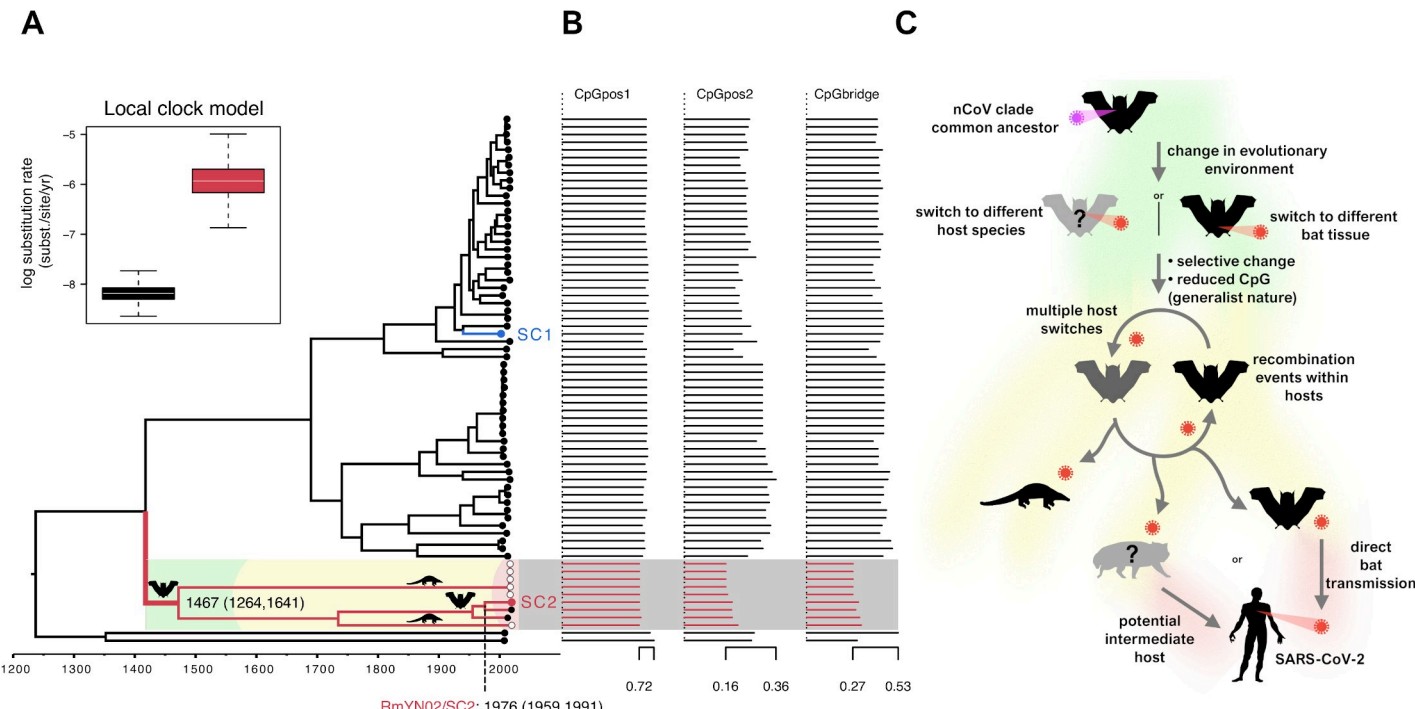

**Fig 3. (A)** Bayesian phylogeny of a modified NRR2 region using different local clocks for the nCoV clade (red branches) and the rest of the phylogeny. All internal nodes' posterior values are above 0.98. Viruses infecting bats are indicated by black circles, pangolins white circles, and SC1 and SC2 labelled in blue and red, respectively, at the tips of the tree. The inset summarises the substitution rate estimates on a natural log scale for the two-parameter local clock model with colours corresponding to the branches in the tree. The estimated date for the shared common ancestor of the nCoV clade (1467) and the RmYN02/SARS-CoV-2 divergence (1976) are shown with confidence intervals. **(B)** CpG relative representation for all dinucleotide frame positions (pos1: first and second codon positions; pos2: second and third codon positions; bridge: third codon position and first position of the next codon) is presented as SDUc values. **(C)** Schematic of our proposed evolutionary history of the nCoV clade and putative events leading to the emergence of SARS-CoV-2. The list of GenBank and GISAID accessions for the *Sarbecovirus* sequences used are provided in S4 Table. nCoV, new coronavirus; SARS-CoV-2, Severe Acute Respiratory Syndrome Coronavirus 2; SC1, SARS-CoV-1; SC2, SARS-CoV-2; SDUc, corrected synonymous dinucleotide usage.

incorporate this substitution rate variation, as they can have a significant impact on estimates (S3 Text).

Zhou and colleagues [57] report a novel bat-infecting *Sarbecovirus* sample, RmYN02, which has the most similar sequence to SARS-CoV-2 of known *Sarbecoviruses* for most of its genome, closer than RaTG13. The part of the RmYN02 Spike ORF that is recombinant falls outside the nCoV clade of the *Sarbecovirus* phylogeny. Molecular dating using BEAST [58] on a region chosen to be a nonrecombinant part of the genome (NRR2) [6] (Fig 3A, see Methods) —accounting for the aforementioned nCoV branch local clock model—indicates that RmYN02 shared an ancestor with SARS-CoV-2 about 1976 (consistent with a previous estimate [29]), and strengthening the evidence for a direct bat to human SARS-CoV-2 emergence. The RmYN02 genome sequence offers an opportunity to test if a recombination event can take place between the 2 distinct CpG content *Sarbecovirus* lineages. A sliding window of CpG relative dinucleotide abundance (RDA) [59] shows that CpG levels of SARS-CoV-2 and RmYN02 only differ at the recombinant region (Fig 4A), further demonstrated by the SDUc values of the nCoV and non-nCoV parts of RmYN02 Spike (Fig 4B). The finding that RmYN02 is a recombinant between the high and low CpG lineages means that viruses from both lineages can coinfect the same bat species. Both the shift in CpG pressures and much of the inference of positive selection at the protein level (Fig 2), coupled with the consistency of the CpG selection pressure throughout the lineage, should reflect a single expansion in evolutionary environment

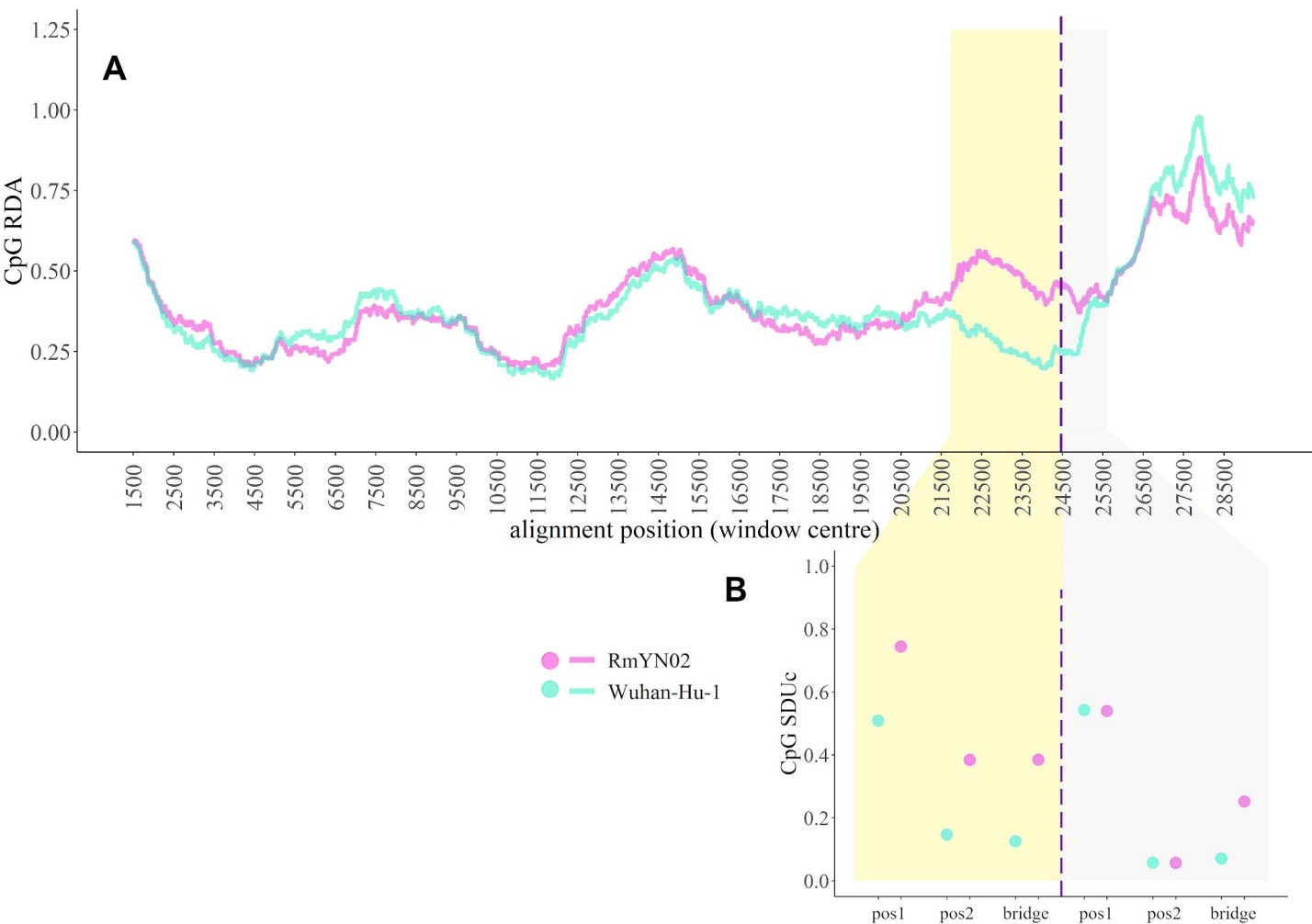

**Fig 4. (A)** 3kb sliding window plot of RDA across the whole-genome alignment of Wuhan-Hu-1 (turquoise) and RmYN02 (magenta). Shaded regions depict the Spike ORF region in the alignment. The dashed line indicates the inferred RmYN02 Spike recombination breakpoint, splitting the shaded region into non-nCoV (yellow) and nCoV (grey). **(B)** SDUc values calculated for each frame position of the 2 RmYN02 Spike nonrecombinant regions and the corresponding Wuhan-Hu-1 regions. The absolute differences between SDUc values of SARS-CoV-2 and RmYN02 for each frame position are significantly greater in the non-nCoV than in the nCoV region ($t_{2.07}$ = 3.03, $p$ = 0.0450; unpaired one-tailed $t$ test with unequal variance). The list of GenBank and GISAID accessions for the Sarbecovirus sequences used are provided in S4 Table. nCoV, new coronavirus; ORF, open reading frame; RDA, relative dinucleotide abundance; SARS-CoV-2, Severe Acute Respiratory Syndrome Coronavirus 2; SDUc, corrected synonymous dinucleotide Usage.

in the lineage leading to the nCoV clade. That viruses from this lineage have not subsequently specialised on novel distinct non-bat hosts is also supported by the absence of related *Sarbecoviruses* in the Sunda/Malayan pangolins prior to their trafficking into China [60]. Together, this suggests the shift at the base of the lineage does not represent specialisation on a host other than bats, but instead an evolved preference towards different host tissues or perhaps circulation in a population of bat species possessing distinct antiviral immunity.

## Discussion

The data presented here highlight that the *Sarbecovirus* clade SARS-CoV-2 emerged from shows evidence of positive selection on its deeper branches, coupled with an adaptive shift in CpG composition in this lineage. Our selection analysis on SARS-CoV-2 genomes across the timeline of the pandemic (Fig 1) indicates a lack of diversifying selection taking place during

the virus's circulation in humans from December 2019 to October 2020. We do find evidence of some ongoing positive selection, presumably associated with minor optimisations to the human population (Fig 1E). However, selection is acting on a small number of sites, and there is a lack of selective sweeps since sequencing began. These results suggest that the majority of adaptive changes which generated SARS-CoV-2 took place prior to its emergence in the human population. There are indications of increased selective pressure in some recent lineages sampled in late 2020, associated with faster spread and higher than usual number of non-synonymous substitutions, for example, the England and South Africa associated lineages B.1.1.7 [61] and B.1.351 [62], respectively. These appear to be associated with SARS-CoV-2 evolution in the context of host immunity due to previous exposure and/or chronic infections of probably immunocompromised individuals (discussed in [63]), and not the slow rate of evolution associated with acute SARS-CoV-2 infections and transmission that has predominated in the pandemic to October 2020.

That SARS-CoV-2 can readily transmit to other animals (pangolins, mink, cats, and others) is strongly indicative this generalist property evolved in the bat reservoir species and not as a consequence of adaptation to human–human transmission. Nonetheless, the amount of time between the first spillover of the progenitor SARS-CoV-2 in humans and sequencing the first variants remains unknown. This raises the concern that important changes might have taken place in that "unsampled" period that cannot be picked up by our SARS-CoV-2 genomic analysis. Despite the inability to directly address these concerns without earlier SARS-CoV-2 sequences or broader sampling of the virus's close relatives, such adaptive changes should theoretically be detected in our *Sarbecovirus* analysis. That we fail to find evidence of diversifying selection on the terminal branch leading up to the emergence of SARS-CoV-2 in humans (Fig 2B), indicates that the adaptations that created a generalist capable of efficient replication in humans and other mammals, probably did not occur in the unsampled SARS-CoV-2 lineage.

Recombination between a proximal ancestor of RmYN02 and a non-nCoV *Sarbecovirus* further demonstrates the viruses in this subgenus cocirculate in the same bat reservoirs. The apparent "success" of these bat viruses to transmit to multiple other mammals and spread with few to no significant genomic changes further supports the hypothesis that the SARS-CoV-2 progenitor is from a viral lineage with a relatively generalist nature (Fig 3C). The level of specialisation of a virus is likely to be driven by many factors, such as mutation rates and available opportunities for transmission, with generalism imposing constraints on change [64]. The recent finding of convergent mutations in mink-circulating SARS-CoV-2 variants, which appear to readily transmit back to humans, demonstrates the potential for low-level specialisation on new hosts following exposure of a generalist virus to a new host species [65]. In fact, our analysis detects some evidence of diversifying selection specific to the terminal branches of the pangolin infecting nCoV viruses (Fig 2B), suggestive of a narrative similar to that in minks. Contrastingly, that is not the case with the terminal branch leading to SARS-CoV-2. Genomes from additional closely related *Sarbecovirsues* would offer additional power to our analyses and provide greater insight into any potential evolutionary hurdles these zoonoses face.

Our analysis suggests that the early ancestors of the nCoV clade—existing hundreds of years ago (Fig 3A and S3 Text)—developed this generalist phenotype through an expansion in their evolutionary environment (host switch, or tissue tropism) within bat species, allowing for efficient spillover events. The shift in CpG suppression at the base of the nCoV clade, which is maintained in all sampled members, could be indicative of an immune evasion adaptation—partly responsible for the proposed generalist properties of the viruses—by evading known CpG-targeting mammalian immune mechanisms such as ZAP [53]. Notably, SARS-CoV-1, the first known *Sarbecovirus* that emerged in humans in recent years is not a member of the nCoV

clade, demonstrating that the nCoV-specific evolutionary changes are not the sole route to a human SARS-like pathogen.

An overarching point stemming from these observations is the lack of sampling and knowledge of the diversity in this viral subgenus. In particular, the closest known bat viruses to SARS-CoV-2 are relatively divergent in time (Fig 3A), and the apparent generalist nature of these viruses suggests that there are species of wild mammals, yet to be sampled, infected with nCoV-like viruses. Serological studies of communities in China that come into contact with bats indicate that incidental and dead-end spillover of SARS-like viruses into humans do occur [66,67]. Due to the high diversity and generalist nature of these *Sarbecoviruses*, a future spillover, potentially coupled with a recombination event with SARS-CoV-2, is possible, and such a "SARS-CoV-3" emergence could be sufficiently divergent to evade either natural or vaccine-acquired immunity, as demonstrated for SARS-CoV-1 versus SARS-CoV-2 [68]. We must therefore dramatically ramp up surveillance for *Sarbecoviruses* at the human–animal interface [15] and monitor carefully for future SARS-CoV emergence in the human population.

## Methods

### SARS-CoV-2 GISAID sequence filtering

After removing any sequence with a nonhuman host (e.g., bat, pangolin), to reduce the impact of sequencing errors on selection analysis, the data from GISAID were filtered by excluding all sequences which meet any of the following criteria: any sequence of length less than 29,000 nucleotides; any sequences with ambiguous nucleotides in excess of 0.5% of the genome; any sequences with greater than 1% divergence from the longest sampled sequence (Wuhan-Hu-1); and any sequence with stop codons. To retain codon alignments and retain correct amino acid sequences for selection analyses, unique (identical sequences are collapsed for alignment to reduce computational expenditure; "N" is treated as matching any resolved character in this process) in-frame nucleotide sequences were translated to amino-acids and aligned with MAFFT [69]. The amino-acid alignment is mapped back to the constituent nucleotide sequences to produce a codon-level alignment. Only unique haplotypes are retained for comparative phylogenetic analyses, since including identical copies is not informative for these types of inference.

### SARS-CoV-2 positive selection

We initially utilised methodology which incorporated the full phylogeny and found misleading signatures such as sequencing errors and lab-based recombination on the terminal branches, which confounded the software (S2 Text). These errors are reflected in the elevated dN/dS ratio observed on the terminal branches (Fig 1). We subsequently used the FEL [30] and MEME methods [36] to infer negative and episodic diversifying positive selection, respectively, using an implementation which only considered internal branches to infer positive selection. MEME uses a maximum likelihood methodology and performs a likelihood ratio test for positive selection on each site, comparing modes which allow or disallow positive diversifying selection at a subset of branches (dN/dS > 1). FEL performs a test that assumes uniform selective pressure on all branches (pervasive selection). These selection analyses were performed in the HyPhy software package v.2.5.14.

### *Sarbecoviruses* alignment and recombination

To avoid the confounding effects of recombination, we have analysed each ORF separately and divided the Orf1ab and Spike ORFs into putative nonrecombinant regions, based on the 7

major recombination breakpoints presented in Boni and colleagues [6]. This produces 5 non-recombinant regions for Orf1ab (regions A to E) and 5 regions for Spike (regions A to D, and the variable loop—region VL). The protein sequences of the nonrecombinant regions SARS-CoV-2, SARS-CoV-1, and 67 closely related viruses with nonhuman hosts (bats and pangolins, S4 Table), identified based on sequence similarity and retrieved from online databases NCBI Genbank and GISAID, were aligned using MAFFT version 7 (L-INS-i) [69]. Subsequent manual corrections were made on the protein alignments and PAL2NAL (http://www.bork.embl.de/pal2nal) was used to convert them to codon alignments. Phylogenies for each codon alignment were inferred using RAxML with a GTR+Γ nucleotide substitution model [70].

## *Sarbecovirus* selection analysis

We used an array of selection detection methods to examine whether the lineage leading to SARS-CoV-2 has experienced episodes of diversifying positive selection. Each nonrecombinant region was examined separately. We separated each region's phylogeny into an nCoV and non-nCoV/SARS-CoV-1 lineage. The nCoV clade includes SARS-CoV-2 and the viruses that form a monophyly with it, excluding the SARS-CoV-1 containing sister clade. These are the bat-infecting viruses CoVZC45, CoVZXC21, RmYN02, and RaTG13, and the pangolin-infecting viruses Pangolin-CoV and the P2V, P5L, P1E, P5E, P4L cluster (S4 Table). Note, some recombinant regions of CoVZC45, CoVZXC21, and RmYN02 do not belong to the nCoV clade, and these were excluded from any analysis of these regions.

We tested for evidence of episodic diversifying selection on the internal branches of the nCoV clade using BUSTED[S], accounting for synonymous rate variation as described in Wisotsky and colleagues [71]. We developed an extension to BUSTED[S], which included an HMM with 3 rate categories to describe site-specific synonymous rate variation [43]. This HMM allows explicit incorporation of autocorrelation in synonymous rates across codons. Such autocorrelation would be expected if selection or mutation rate variation were spatially localised within ORFs. The rate switching parameter between adjacent codons of the HMM describes the extent of autocorrelation, with values under 1/N (N = number of rate classes) suggestive of autocorrelation. Standard HMM techniques (e.g., the Viterbi path) applied to these models can reveal where the switches between different rate types occur, thereby partitioning the sequence into regions of weaker or stronger constraint on synonymous substitutions.

The aBSREL method [41] was used on all branches of the nCoV clade to determine which specific branches drive the inference of selection. Finally, we examined which specific codon sites are under negative selection on average over the nCoV clade using FEL [30] and under pervasive or episodic diversifying positive selection on the nCoV clade using MEME [36]. $P$ values of ≤0.05 for the likelihood ratio tests, specific to each method, were taken as evidence of statistical significance. Most selection analyses were performed in the HyPhy software package v.2.5.14, with BUSTED adjusted to allow multiple nucleotide substitutions using v2.5.24 [72].

## CpG depletion

To quantify over/under representation of CpG dinucleotides in the *Sarbecovirus* genomes, we developed a modified version of the synonymous dinucleotide usage (SDU) metric [55], which now accounts for biased base composition. The original SDU metric compares the observed proportion of synonymous CpG, $o$; for each pair of frame positions in a coding sequence, $h$; to that expected under equal synonymous codon usage, $e$; for each amino acid (or amino acid pair) that can have CpG containing codons (or codon pairs), $i$. The SDU metric is the mean of

these ratios weighted by the number of informative amino acids (or pairs) in the sequence, *n* (Eq 1).

To incorporate the biased and variable base composition of SARS-CoV-2 and other *Sarbecoviruses* [47], here we have estimated expected codon usage based on each virus' whole-genome nucleotide composition. We term this new metric the corrected synonymous dinucleotide usage (SDUc). We use observed base frequencies from each virus to generate the corrected null expectation of the metric, *e′*, instead of assuming equal usage (Eq 1). The expected proportion, *e′*, for every amino acid/amino acid pair was estimated by randomly simulating codons based on the whole-genome single nucleotide proportions of each virus. This *e′* was then used for all SDUc calculations of the corresponding virus.

As this metric is susceptible to error when used for short coding sequences, we applied SDUc on the longest ORF (Orf1ab) of all the viruses. To estimate the extent of phylogenetic independence between synonymous sites across SDUc datapoints, we measured the pairwise synonymous divergence (Ks) between viruses. Pairwise Ks values were calculated using the seqinr R package [73], which utilises the codon model of Li (1993) [74], demonstrating the partial but not complete independence within the 2 lineages. The Ks median and maximum is 0.54 and 0.89 within the nCoV clade, and 0.34 and 1.09, respectively, within the non-nCoV/SARS-CoV-1 clade.

$$SDUc_{CpG,h} = \frac{\sum_{i=1}^{k} n_i \times \dfrac{o_{i,h}}{e'_{i,h}}}{N}$$

Eq 1

(Eq 1, N = total number of informative amino acids)

## Spike recombination analysis

To determine the recombination breakpoint on the Spike ORF of the RmYN02 virus, we used the RDP5 method suite [75], implementing 7 methods: RDP, GENECONV, Chimaera, Max-Chi, BootScan, SiScan, and 3seq. We first performed the analysis on the whole-genome alignment of the *Sarbecoviruses* and then determined the relevant breakpoint within the Spike ORF by rerunning the method on the Spike-only alignment. The accepted breakpoint (position 24058 in the RmYN02 genome) was consistently called by 6 out of the 7 tested methods (RDP, GENECONV, Maxchi, Chimaeara, SiSscan, and 3seq). Similarly, the preceding breakpoint of the non-nCoV region was called at position 21248 of the RmYN02 genome (before the start of the Spike ORF).

## Bayesian inference under a local molecular clock

To assess the overall nonrecombinant phylogenetic relation between the viruses, we used the longest nonrecombinant genomic regions described in Boni and colleagues [6], NRR1 and NRR2, and added RmYN02 to the alignments. Based on the recombination breakpoints determined above for RmYN02, NRR2 was adjusted to end at position 21266 of Wuhan-Hu-1, instead of 21753, corresponding to the start of the RmYN02 recombinant region (RmYN02 position 21248). Time-measured evolutionary histories for NRR1 and NRR2 were inferred using a Bayesian approach, implemented through the Markov chain Monte Carlo (MCMC) framework available in BEAST 1.10 [58]. Motivated by the observation of a higher root-to-tip divergence for the nCoV clade (see S3 Text and S5 Fig) and by the CpG trait shift on the branch ancestral to the nCoV clade (see below), we specified a fixed local clock [76] that allows for a different rate on the branch leading to the nCoV clade. In the absence of strong temporal signal, we specified an informative normal prior distribution (with mean = 0.0005 and

standard deviation = 0.0002) on the rate on all other branches based on recent estimates under a relaxed molecular clock [77]. To maintain the overall topology under the local clock model, we constrained the Kenyan (KY352407) bat virus as outgroup for the viruses from China in NRR1, and the Kenyan as well as a Bulgarian bat virus (NC_014470) as outgroup for the viruses from China in NRR2. We partitioned the coding regions of NRR1 and NNR2 by codon position and specified an independent general time-reversible (GTR) substitution model with gamma distribution rate variation among sites for each of the 3 partitions. We used a constant size coalescent model as tree prior and specified a lognormal prior with mean = 6.0 and standard deviation = 0.5 on the population size. Three independent MCMC analyses were run for 250 million states for each data set. We used the BEAGLE library v3 [78] to increase computational performance. For completeness, we performed the same analysis specifying the local clock model on the entire nCoV clade instead of just the branch leading to it. This has an effect on the substitution rate and node time estimates; however, we cannot formally distinguish which model is a better fit (S3 Text). We focused on the branch-only local clock model, since it directly tests our hypothesis that the rate change is linked to the CpG adaptive shift specific to that branch. Fig 3A presents the phylogeny for NRR2 because it is the longest intact nonrecombinant region and could possibly yield more reliable estimates. The additional NRR1 phylogenies and nCoV clade local model phylogenies are presented in S3 Text. BEAST parameter XML files are provided in S1 Data (sequence alignments are excluded to abide to GISAID data sharing restrictions). Continuous parameters were summarised and effective sample sizes were estimated using Tracer [79]. Divergence time estimates for the nCoV clade are summarised in S3 Text. Trees were summarised as maximum clade credibility (MCC) trees using TreeAnnotator and visualized using FigTree (http://tree.bio.ed.ac.uk/software/figtree/).

## Identifying shifts in CpG content

Shifts in CpG content were identified using a phylogenetic comparative method that infers adaptive shifts in multivariate correlated traits with the R package PhylogeneticEM [56]. This approach models trait evolution on phylogenies using an Ornstein–Uhlenbeck (OU) process and uses a computationally tractable version of the full multivariate OU model (scalar OU) for multivariate traits. Estimates of the shift positions are obtained using an Expectation–Maximization (EM) algorithm. The shift positions are estimated for various numbers of unknown shifts, and a lasso-regression model selection procedure identifies the optimal number of shifts. We applied the procedure to the MCC trees for NRR1 and NRR2 with their respective CpG SDUc values (ln transformed as required by the phylogenetic approach). The trees were forced to be ultrametric, as required for the scalar OU model, by extending all the external edges of the trees to match the most recently sampled tip. Using this procedure, we identified 3 and 2 CpG shifts in both NRR1 and NRR2, respectively (S6 Fig), with the shift on the branch ancestral to the nCoV clade being the only consistent one identified in both genomic regions (Fig 3A).

## Supporting information

**S1 Text. Inferring population demography from distribution of allele frequencies.**
(PDF)

**S2 Text. Signals of positive selection in SARS-CoV-2 and frequency-based analysis of SARS-CoV-2 polymorphisms.**
(PDF)

**S3 Text. Model selection when testing for a nCoV-specific rate change.**
(PDF)

**S1 Fig. Phylogenetic tree (generated with RAxML and visualized using FigTree), showing the relationship of SARS-CoV-1 and SARS-CoV-2 (orange) to related bat and pangolin _Sarbecoviruses_ from the whole genome alignment.** Grey, red, and blue variants are coloured according to Letko and colleagues [12] who showed experimentally some viruses are able to use human ACE2 (red), while some require exogenous protease treatment in vitro (grey); the red outlier is a known recombinant. Black indicates not tested by Letko and colleagues, while the virus in blue (sampled in Bulgaria) could not be induced to infect human cells. The scale bar corresponds to nucleotide substitutions per site.
(PNG)

**S2 Fig.** Schematic representation of selection methods implemented on each data set of the analysis: (i) the 133,741 SARS-CoV-2 genomes and (ii) the 69 Sarbecovirus genomes.
(PNG)

**S3 Fig. Synonymous rate of each of the 3 synonymous rate variation classes estimated by BUSTED[S] + HMM and the proportion attributed to each class for all nonrecombinant open reading frames.**
(PNG)

**S4 Fig.** Corrected synonymous dinucleotide usage (SDUc) values for the Orf1ab of each Sarbecovirus for all dinucleotide frame positions: (A) pos1: first and second codon positions, (B) pos2: second and third codon positions, and (C) bridge: third codon position and first position of the next codon, plotted against patristic distance from SARS-CoV-2 (reference genome Wuhan-Hu-1). (D) The tip colours of the phylogeny correspond to the SDUc data points. SARS-CoV-1 is labelled in orange in panels A, B, and C for comparison. The non-nCoV part of the phylogeny has been collapsed for clarity.
(PNG)

**S5 Fig. Maximum likelihood trees for NRR1 (left) and NRR2 (right).** The trees were inferred using IQTREE using a GTR substitution model with gamma-distributed rate variation among sites. The nCoV lineage is indicated in red. SARS and SARS-CoV-2 are shaded in blue and red, respectively.
(PNG)

**S6 Fig. Estimates of shifts in CpG in the _Sarbecovirus_ phylogeny based on NRR1 (left) and NRR2 (right).** The log-transformed CpG content values are shown at the tips of the trees. The identified shifts are indicated with black circles on their respective branches and with different colours for the lineages and CpG measures involved. The inset shows the results for penalized least-squares model selection criterion (BGHml).
(PNG)

**S1 Table. List of GISAID accessions used for FUBAR analysis (see S2 Text).**
(CSV)

**S2 Table. List of sites found to be under selection by MEME in the nCoV clade.**
(XLSX)

**S3 Table. BUSTED[S] fits for nonrecombinant segments.** For each segment, 2 rows of numbers are presented: top—for the BUSTED[S]-HMM model and bottom—for the BUSTED[S] model, allowing for multinucleotide substitutions (see Lucaci and colleagues [45] for details). The model with the better AIC-c and significant test results ($p$-value) are shown in bold. CV synonymous rate variation (SRV): coefficient of variation (CV) of the site-to-site synonymous rate distribution. $\lambda$: the rate of double-nucleotide substitutions (relative to the rate of single-

nucleotide synonymous substitutions); δ: the relative rate of triple nucleotide substitutions; δ$_{islands}$: the relative rate of triple nucleotide substitutions that involve synonymous codons, e.g., for Serine.
(XLSX)

**S4 Table. List of *Sarbecoviruses* used in the selection analysis and their GISAID accessions.**
(CSV)

**S5 Table. GISAID accessions used in the SARS-CoV-2 selection analysis.**
(CSV)

**S6 Table. GISAID sequences acknowledgement table.**
(PDF)

**S1 Data. XML files with BEAST model parameters (sequences are excluded due to GISAID data sharing restrictions).**
(GZ)

## Acknowledgments

We would like to thank all the authors who have kindly deposited and shared genome data. A table with GISAID genome sequence acknowledgments can be found in S6 Table. Credit also needs to be given to the surveillance projects for generating the genome data that are available in GenBank and to the software developers for making the tools we have used freely available. We thank Alex Gunnarsson, Xiaowei Jiang, Joseph Hughes, and Kyriaki Nomikou for thankful comments on the manuscript.

## Author Contributions

**Conceptualization:** Oscar A. MacLean, Spyros Lytras, Sergei L. Kosakovsky Pond, David L. Robertson.

**Data curation:** Spyros Lytras, Joshua B. Singer, Maciej F. Boni.

**Formal analysis:** Oscar A. MacLean, Spyros Lytras, Steven Weaver, Philippe Lemey, Sergei L. Kosakovsky Pond, David L. Robertson.

**Funding acquisition:** Maciej F. Boni, Philippe Lemey, Sergei L. Kosakovsky Pond, David L. Robertson.

**Investigation:** Oscar A. MacLean, Spyros Lytras, Steven Weaver, Philippe Lemey, Sergei L. Kosakovsky Pond, David L. Robertson.

**Methodology:** Oscar A. MacLean, Spyros Lytras, Maciej F. Boni, Sergei L. Kosakovsky Pond.

**Resources:** Steven Weaver, Joshua B. Singer, Sergei L. Kosakovsky Pond.

**Software:** Steven Weaver, Joshua B. Singer, Sergei L. Kosakovsky Pond.

**Supervision:** Sergei L. Kosakovsky Pond, David L. Robertson.

**Visualization:** Spyros Lytras, Philippe Lemey, Sergei L. Kosakovsky Pond.

**Writing – original draft:** Oscar A. MacLean, Spyros Lytras, Philippe Lemey, Sergei L. Kosakovsky Pond, David L. Robertson.

**Writing – review & editing:** Oscar A. MacLean, Spyros Lytras, Maciej F. Boni, Philippe Lemey, Sergei L. Kosakovsky Pond, David L. Robertson.

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
