## [Editor Report · Decision Letter 0]

23 Oct 2020

Dear Dr. Robertson, 

Thank you for submitting your manuscript entitled "Natural selection in the evolution of SARS-CoV-2 in bats, not humans, created a highly capable human pathogen" for consideration as a Research Article by PLOS Biology.

Your manuscript has now been evaluated by the PLOS Biology editorial staff and I am writing to let you know that we would like to send your submission out for external peer review.

Please re-submit your manuscript within two working days, i.e. by Oct 25 2020 11:59PM.

Kind regards,

Paula

---

associate Editor

PLOS Biology

---

## [Decision Letter · Decision Letter 1]

4 Dec 2020

Dear Dr. Robertson,

Thank you very much for submitting your manuscript "Natural selection in the evolution of SARS-CoV-2 in bats, not humans, created a highly capable human pathogen" for consideration as a Research Article at PLOS Biology. Your manuscript has been evaluated by the PLOS Biology editors, an Academic Editor with relevant expertise, and by several independent reviewers.

In light of the reviews (below), we are pleased to offer you the opportunity to address the comments from the reviewers in a revised version that we anticipate should not take you very long. We will then assess your revised manuscript and your response to the reviewers' comments and we may consult the reviewers again.

In particular, referee #1 says that due to the under sampling in wildlife species, a greater sampling of animal species in nature including bats could indicate a closer ancestor and may well demonstrate selection and host-adaptations, and says that you should recognize this and the paucity of SARS-CoV-2 genomes from very early in the pandemic from China in the text, adding that you can’t dismiss the hypothesis of human-specific adaptation. Referee #1 also has several questions and comments about your analysis and assumptions, that we consider all should be addressed. Referee #2 has suggestions for modifying text and figures in order to make it clearer for a general audience. Both reviewers think that you should change the title.

We expect to receive your revised manuscript within 1 month.

**IMPORTANT - SUBMITTING YOUR REVISION**

*Resubmission Checklist*

*Published Peer Review*

*PLOS Data Policy*

*Blot and Gel Data Policy*

Sincerely,

Paula

---

Associate Editor,

pjaureguionieva@plos.org,

PLOS Biology

REVIEWER'S EXPERTISE:

Reviewer #1: viral phylodynamics

Reviewer #2: dynamics of zoonotic infections

Reviewer #1: This manuscript by MacLean/Lytras and colleagues is a fascinating analysis assessing the nature of selection occurring in horseshoe bats compared to SARS-CoV-2 evolution in humans. In this study the authors explore the evolutionary history of bat Sarbecoviruses that may help shed some insight into the emergence and rapid spread of SARS-CoV-2. One of the quirks of this pandemic is that the virus responsible evolves relatively slowly resulting in a very shallow phylogenetic tree. This feature inhibits the statistical power of standard comparative methods and as such other computational approaches are needed. This manuscript adapts a rigorous methodology to measure the strength and direction of selection from SARS-CoV-2 and its Bat precursors while carefully considering confounding errors (sequencing or other lab-based errors) that may have occurred on terminal branches. 

One thing that does irk me slightly is the authors assertions of the "generalist" nature of these viruses allowing for efficient spillover events. While there is no doubt that SARS-CoV-2 is well adapted for humans and the authors have not found any evidence for selection in the ancestral branches leading to SARS-CoV-2 there is a huge amount of under sampling in wildlife species so a greater sampling of animal species in nature including bats could indicate a closer ancestor and may well demonstrate selection and host-adaptations. This caveat and the paucity of SARS-CoV-2 genomes from very early in the pandemic from China should be duly recognized within the text. Moreover, given the divergence estimates from RmYN02/SARS-CoV-2 of 1976 there is potentially decades of unobserved evolution that may have occurred that warrants consideration. 

Specific comments.

1. The methods are quite technical and for a broader reader of PLoS Biology I think they would find it challenging to understand the nuances of each computational approach. It may be worth the authors considering a figure (even as supplementary) to highlight the methodological approach used.

2. The authors premise that the majority of host adaptations occurred before the emergence of SARS-CoV-2 in humans while may be technically sound with the current sequence data but how can the authors dismiss the hypothesis that human-specific adaptation would have likely reached fixation even before the first SARS-CoV-2 genome was sequenced?

3. As this field is continuously evolving with new genomic data being added daily the statement in line 328-331 should no longer be considered accurate as we can now observe rapid adaptation of SARS-CoV-2 in mink populations. E.g Y453F in the receptor biding motif and this rarely occurs in humans. My point linked with the above point is that increased surveillance permitted us to observe this variant in minks while similarly changes may have occurred earlier in the pandemic and become fixed before sequencing was done. Could rapid adaptation in late 2019 from unsampled asymptomatic transmission chains be plausible? 

4. Unsurprisingly, the authors find that up until early June, relatively weak purifying selection was acting on SARS-CoV-2 sequences. If the authors extended their analysis to the present what would they expect to find given that there is more diversity in current circulating variants? 

5. While SARS-CoV-2 consensus sequences remain largely unchanged over time within hosts is there evidence for a more dynamic evolutionary process sub-consensus? 

6. The authors use MEME as documented in Figure 1c to individual sites subjected to episodic diversifying selection. However, I thought that MEME assumes that selective pressures between branches are uncorrelated. Surely, this is not the case for SARS-CoV-2 as changes are occurring very slowly across the phylogeny as neighboring branches will be correlated. Have the authors considered a mixed effects covarion model and if this would improve power to detect directional selection?

7. The addition of synonymous site rate variation is a great addition to BUSTED as constant dS rates can elevate false-positive and reduce power to test individual sites for selection. However, can the authors comment on whether accommodating synonymous rate variation results in reduced power compared to the original method? 

8. How did the authors consider multinucleotide mutations in their analysis as I am sure the authors are aware that there is a high possibility of false positives with branch sites tests like BUSTED. For example, Venkat et al. (2018). 

9. From figure 2 there appears to be evidence for selection in the pangolin CoV cluster within Orf1ab. While not the focus of this paper does this not suggest that there may be other adaptations under selection and responsible for its emergence in this animal host? 

10. The title while scientifically accurate is a bit clumsy to me. I would suggest something catchier for a reader. 

11. How did the authors consider genomes with Ns in collapsing sequences into unique haplotypes. Were they ignored and only A,C,G,T characters considered? 

Reviewer #2: This important paper explores the evolutionary selective pressure on the Sarbecovirus subgenus of viruses and a subset of early SARS-CoV-2 genomes circulating in the human population in order to assess whether natural selection facilitated SARS-CoV-2's cross-species transmission to and consequential spread in the human population. In particular, the authors undertake the following key analyses:

1. Exploration of natural selection in SARS-CoV-2 (hereafter, SC2 for simplicity) human genomes. 

First, the authors analyze, ~50000 human genomes of SC2, up until June 28, 2020 of the pandemic, limiting their analysis to genomes >29000 bps, with <1% divergence from the reference, with <0.5% ambiguous bases, and lacking stop codons. They use this dataset to assess evidence of purifying and/or positive selection in SC2, finding that most genetic variants in the virus occur at low frequency (in <15 genomes) and exhibit weakly purifying selection, consistent with a model of exponential virus growth. A few exceptions occur in the case of a few high frequency variants found in >3000 genomes that show weakly positive selection. 

The authors undertake a number of advanced analyses to validate these few SNPs that are deemed to be under positive selection, identifying 10 candidate mutations with dN/dS ratios >1 that could be positively selected. They consider each of these individually, investigating the timing and laboratory of submission, the possibility of sequencing error or recombination to give rise to this variant and ultimately converge on four mutations that appear to be truly under positive selection: RdRp 323, S943, S614, and S141.

2. Analysis of selection in SARSr-CoV sequences in bats.

For the second major analysis of the paper, the authors analyze a subset of 19 non-recombinant regions of several bat/human/pangolin SARSr-CoVs identified in Boni et al 2019. Each non-recombinant region is analyzed independently, and for each region, the authors separate the viruses into an nCoV clade, representing those most closely related to SC2 and forming a monophyly and a non-nCoV clade, representing those more distantly related. 

Using the program aBSREL, the authors first search for positive selection on specific branches in the SARSr-CoV phylogeny and find its imprints in the deepest branches of the nCoV lineage long before the emergence of the virus to humans, suggesting that it was not recent selection that allowed for the cross species shift.

The authors next use a program called BUSTED and their own extension to explore synonymous rate variation (SRV) across SARSr-CoV genomes, finding positive selection in the nCoV clade in Orf1ab, Spike and N proteins. Using the MEME method, the authors identify 85 particular sites in the nCoV clade under selection, most in Orf1ab, Spike, and N. Critically, they show a higher than expected proportion of sites in Orf3a, for which the function is not known, but they suggest it might play a role in immune evasion.

The authors' extension to the BUSTED method allowed them to infer differing substitution rate classes across the non-recombinant sites in the genome; the authors determine up to 200-fold differences in the rate of synonymous substitution across sites in the genome, suggesting that some synonymous sites may be under strong purifying selection to purge deleterious mutations.

3. CpG depletion in the nCoV clade

The third and final major analysis of the paper investigates the depletion of CpG sites in the nCoV clade. CpG depletion is believed to be advantageous for virus evolution because It aids in evasions of a CpG-targeted mammalian immune response involving Zinc-finger Antiviral Protein (ZAP), as well as antiviral C to U hypermutation carried out by APOBEC3 cytidine deaminases. The authors use a framework called Synonymous Dinucleotide Usage (SDUc) to compare CpG representation across the 19 non-recombinant regions of the SARSr-CoV clade, finding significant CpG under-representation in Orf1ab for all Sarbecoviruses and lower CpG content overall in the nCoV clade Sabecoviruses vs all others. They fit this trait on two alignments of the 19 non-recombinant regions of the SARSr-CoVs to identify points where the "CpG suppressive" trait evolved: in particular, on the lineage leading to the nCoV clade. 

Finally, they tested a model which allowed for a relaxed mutation rate in different clades to find evidence of an elevated substitution rate on the nCoV lineage subsequent to this CpG depletion event, giving way to a generalist virus clade.

General comments: 

Though familiar with the SC2 evolutionary history literature, I am not a phylogeneticist by training and cannot comment critically on the methods selected for recombination and selection analysis (aBSREAL, MEME, BUSTED, SDUc). These appear to be appropriate to me. More generally, however, I believe that this paper is an important and relevant contribution to the SC2 evolution literature and should be published soon in PLoS Biology. It is not, however, the most clearly written paper I have encountered, and I have a few suggestions for how the authors could make their findings more accessible. 

Title: Title is a bit of a mouthful. What about dropping the "not humans" bit and including the word "generalist" -- something to the effect of: "Natural selection of SARS-CoV-2 in bats created a generalist virus and highly capable human pathogen"

Intro: It would be helpful if you set up these three major analyses summarized above at the end of the intro (~Line 99). Because PLoS Bio requires results to come before Methods, it can often lead to slightly disjointed papers, but I think one quick sentence preparing the reader for the three major areas of focus to follow would help a lot. In addition, I would recommend being very explicit about the two datasets used in this paper and stating that analysis #1 refers to the 50000 human SC2 genomes while analyses #2 and 3 refer to the SARSr-CoV dataset (in fact, two versions of it as presented in Boni et al. 2020)

Discussion: I think some discussion of the pros vs. cons of being a specialist vs. a generalist virus is warranted here. The paper suggests that the entire clade of Sarbecoviruses is a highly generalist clade which seems like it should be a majorly adaptive feature. Why then have they not outcompeted all of the other tradeoffs? A nod to the literature on specialism vs. generalism in host-pathogen coevolution would be appropriate.

Figures

Fig 1: Fig 1 is largely appropriate for summarizing the analysis of human SC2 genomes and evidence of positive selection but Fig 1A could be improved: it shows 'Variant Frequency' on the x-axis, which I believe gives the number of genomes in the dataset for in which a given variant is found, by both the dN/dS ratio on the primary y-axis (dots) and the count of variants of this type in the dataset on the secondary y-axis. So I think it is saying that there are ~7000 individual variants which are unique to only one genome in the dataset and these have a mean corresponding dN/dS ratio around .85. Likewise, I see it as saying that there are <50 variants that are found in over 3000 genomes and that these have an average dN/dS ratio of around 1.4. Is this a correct interpretation? 

It would be helpful if the bar color marched the color of the secondary y-axis text (count) and the dots matched the color of the primary y-axis text (dN/DS) to avoid confusion.

Also, why is there no range of error for the bars (i.e. through bootstrapping) or the dots from the SLAC method?

Fig 2. I found Fig 2 difficult to interpret. I have a few suggestions for how it could be improved: 1. Break it down into subsections (A, B, C) so that you can refer to each separately in the caption.

2. In the caption, explain the genome structure and positive/negative selection subset first, sinc these are at the top of the figure (and refer to this as figure component A). 

3. The individual phylogenies are fine—just group these together as part B. I *think* the color scale for dN/dS ratio corresponds to the mean ratio inferred for the highlighted clades within the lineage, so if these are subgrouped together it will be easier to understand.

4. Then, discard the donut plots. It is unclear why they are only present for some of the ORF regions (presumably only those with significant positive selection are shown?) and also confusing that their color scheme is different from the phylogenies. Instead, just either refer to the supplementary table as is done anyway or make a small table as part C that lists omega3 parameter for each ORF from Table S4 and corresponding percentage and mention in the caption that omega1 and 2 were basically 0 for all parameters.

Fig 3. Lovely figure. Could be slightly easier to explain if you made the CpG values part B of the figure and the schematic part C (or if the schematic were a different figure entirely—they don't really relate to one another).

Fig 4. In general, very clear. Can you do some comparative stats on these SDUc values for the frame positions in these two regions of the spike protein (inset) to show that WuHan and RmYN02 don't differ in the nCoV half but do in the non-nCoV half?

A few minor line-by-line comments here:

Line 43: change to "which created a relatively generalist"

Line 47-50: wording is awkward. Change to "Evolutionary analysis identified this new virus to humans as a Severe acute respiratory syndrome-related coronavirus [1], in the Sarbecovirus subgenus of the Betacoronavirus genus, sister to the original SARS virus; it was subsequently named SARS-CoV-2 to reflect this relationship [2]."

Line 62-63: it is not proven that SARS-CoV transmitted to humans via civets and the cited ref is just a review paper. Suggestion to change to "Later it became clear that while these animals may have been conduits for spillover to humans, they were not true viral reservoirs"

Line 88: suggestion to change "us" to "humans"

Line 144: would be good to here cite Plante et al 2020, now published in Nature

Line 242: should be one sentence

Line 359: why was 1% chosen as a cutoff for too divergent (I agree, as 1% divergence would far outpace the known mutation rate for SARS-CoV-2 but I think you should provide a ref indicating the reasonable range of expected divergence, especially up to the point in time studies in your data subset)

Line 360: sudden shift to first person, present tense is perplexing. Please keep tense consistent throughout the paper. Past tense probably makes more sense (i.e. "the data from GISAID was filtered").

Supplementary Materials:

Table S4 could be easily incorporated into the pdf file for the supplementary materials and would be more accessible that way.

---

## [Editor Report · Decision Letter 2]

15 Jan 2021

Dear Dr. Robertson,

Thank you for submitting your revised Research Article entitled "Natural selection in the evolution of SARS-CoV-2 in bats created a generalist virus and highly capable human pathogen" for publication in PLOS Biology. I have now obtained advice and discussed with the Academic Editor. 

We will probably accept this manuscript for publication, assuming that you will modify the manuscript to address the editorial requirements. Please also make sure to address the data and other policy-related requests noted at the end of this email.

In particular, it seems that you provide data for supplementary figures 1, 3, 6 and 7 but it is not stated where the data can be found and it is not very clear to what figure this data belongs to. Please see at the end of this email for more information.

We expect to receive your revised manuscript within two weeks. Your revisions should address the specific points made by each reviewer.

-  a cover letter that should detail your responses to any editorial requests, if applicable

*Published Peer Review History*

*Early Version*

Sincerely,

Paula

---

Associate Editor,

pjaureguionieva@plos.org,

PLOS Biology

DATA POLICY:

Regardless of the method selected, please ensure that you provide the individual numerical values that underlie the summary data displayed in the following figure panels as they are essential for readers to assess your analysis and to reproduce it: Figure 1A, 1B, 1C, 1D, 1E, 2B, 3A, 4A, 4B, Supplementary figure 8A, 8B, 8C, 8D, Supplementary figure 10, Supplementary figure 11, Supplementary figure 12.

Please also **ensure that figure legends in your manuscript include information on where the underlying data can be found, and ensure your supplemental data file/s has a legend.**

Please **ensure that your Data Statement in the submission system accurately describes where your data can be found.**

---

## [Editor Report · Decision Letter 3]

25 Jan 2021

Dear David,

On behalf of my colleagues and the Academic Editor, Damien Tully, I am pleased to say that we can in principle offer to publish your Research Article "Natural selection in the evolution of SARS-CoV-2 in bats created a generalist virus and highly capable human pathogen" in PLOS Biology, provided you address any remaining formatting and reporting issues if needed. These will be detailed in an email that will follow this letter and that you will usually receive within 2-3 business days, during which time no action is required from you. Please note that we will not be able to formally accept your manuscript and schedule it for publication until you have made the required changes.

Thank you again for supporting Open Access publishing. We look forward to publishing your paper in PLOS Biology. 

Sincerely, 

Paula

---

Paula Jauregui, PhD 

Associate Editor 

PLOS Biology
